



# Computation of covariant lyapunov vectors using data assimilation

Shashank Kumar Roy[1] and Amit Apte[2]

[1,2]International Centre for Theoretical Sciences, Bangalore 560064 India
[2]Indian Institute of Sciences, Education and Research, Pune 411008 India

**Correspondence:** Shashank Kumar Roy (shashank.roy@icts.res.in)

**Abstract.** Computing Lyapunov vectors from partial and noisy observations is a challenging problem. We propose a method using data assimilation to approximate the Lyapunov vectors using the estimate of the underlying trajectory obtained from the filter mean. We then extensively study the sensitivity of these approximate Lyapunov vectors and the corresponding Oseledets' subspaces to the perturbations in the underlying true trajectory. We demonstrate that this sensitivity is consistent with and helps explain the errors in the approximate Lyapunov vectors from the estimated trajectory of the filter. Using the idea of principal angles, we demonstrate that the Oseledets' subspaces defined by the LVs computed from the approximate trajectory are less sensitive than the individual vectors.

## 1 Introduction

The earth system is a highly nonlinear, chaotic, complex dynamical system with a very large number of degrees of freedom. Predictability of such a system, for example for the purposes of numerical weather prediction, is limited due to various factors, including uncertainties in the dynamical models as well as uncertainties in initial conditions which grow exponentially due to the inherent chaotic nature of the dynamics (Kalnay, 2003). Thus predictions of the state of the system beyond a certain time, related to the Lyapunov time scale, may become so uncertain as to become irrelevant. In order to keep the state as represented by a numerical model close to reality over time, we need to use observations, which are always noisy and usually sparse both in space and time. The process where observations and the model dynamics are "combined" to estimate the model state or its probability distribution that is close to the true state under some metric is broadly called data assimilation (DA).

Grounded in the principles of filtering theory, data assimilation is a well-known technique in geosciences for state estimation problems (See, e.g., Kalnay, 2003; Law et al., 2015; Reich and Cotter, 2015; Van Leeuwen et al., 2015; Asch et al., 2016; Fletcher, 2017; Carrassi et al., 2018, for reviews and further references). Some of the most commonly used methods are based on Monte Carlo approximations of the conditional distribution, called the posterior or the filter, of the state conditioned on observations. The mean and the covariance is used commonly as estimates of the state and associated uncertainty. We use the popular Monte Carlo method called ensemble Kalman filter in this work (Evensen, 2003).

The uncertainties in the state estimates, obtained using DA, depend quite crucially on the directions of the error growth, or in other words, the dynamical instabilities of the system. Lyapunov exponents and vectors are the fundamental tools used in the study of nonlinear and chaotic systems, in order to characterize the stability properties of a dynamical system with respect to perturbations along different directions. The paradigm called assimilation in unstable subspace (see, e.g., Trevisan and Uboldi,



2004; Carrassi et al., 2007, 2008a, b; Trevisan et al., 2010) uses the subspace spanned by the unstable and neutral Lyapunov vectors for producing the analysis or the update step of assimilation and shows promising improvements over the traditional algorithms. The importance of Lyapunov vectors in general and in the context of assimilation is also discussed extensively in recent works of Palatella et al. (2013); Vannitsem and Lucarini (2016); Bocquet et al. (2017); Gurumoorthy et al. (2017); Brugnago et al. (2020) and others.

For a dynamical systems, a non-increasing tuple $\lambda_1 > \lambda_2 > ... > \lambda_s$ of Lyapunov exponents summarizes the global, asymptotic rate of change of linear perturbations around a trajectory. Existence of at least one positive Lyapunov exponent indicates exponential divergence of perturbations and is indicative of instabilities and chaotic dynamics. The associated directions in the tangent space are called Lyapunov vectors which span the tangent space at a specific point in the phase space and contain the information about the past and future evolution of local perturbations (Eckmann and Ruelle, 1985; Legras and Vautard, 1996). Various methods have been proposed for computation of Lyapunov exponent, with or without the use of the dynamical equations and / or their linearization, or using a time series of observations of the system (Benettin et al., 1980b, a; Abarbanel, 1996).

Lyapunov vectors that capture the asymptotic growth rate as $t \to \pm\infty$ are called, respectively, forward or backward Lyapunov vectors (FLV / BLV). These FLV and BLV provide a orthonormal basis for a filtration of the tangent space, but they are not covariant with respect to the dynamics: the time evolution, under the linear dynamics, of an FLV (resp. BLV) at one time does not lead to an FLV (resp. BLV) at other time but leads to a linear combination of FLVs (resp. BLVs). Further, even though the asymptotic growth rate of the FLV in forward time is the Lyapunov exponent, it is not so in backward time (and similarly for BLV). The vectors that have both these properties, namely covariance with respect to the time evolution and asymptotic growth rates in forward and backward time, are called covariant Lyapunov vectors (CLV), which are the central focus of this work. In recent years, various methods for computing these Lyapunov vectors have been developed (Wolfe and Samelson, 2007; Kuptsov and Parlitz, 2012; Ginelli et al., 2013; Noethen, 2019).

The main aims of this paper are twofold - one is to present a data-based algorithm for computation of Lyapunov vectors and the other is to study their sensitivity to noise.

1. Firstly, we present (in section 2.3) a data-based algorithm for computing the LVs using the state estimates obtained from a data assimilation method, namely the ensemble Kalman filter (though any assimilation method may be used in the place of EnKF). This algorithm can be used to produce the full spectrum or a subset of LVs, either backward or covariant. This data-based method does not, by itself, give any bounds on how close these vectors may be to the true vectors associated with the trajectory that is being observed. In order to understand this aspect, we are naturally led to the second aim of this paper.

2. Secondly, we present (in section 4) the other main contribution which is an extensive numerical exploration of the sensitivity of the LVs to perturbations of the underlying trajectory. We do this by using the same algorithm mentioned above but with a noisy trajectory and then comparing the true LVs with the approximate one obtained from the perturbed trajectory. We use the principle subspace angles between the Oseledets spaces in order to quantify this discrepancy.



The first part - a data-based algorithm for calculating CLV presented in 2.3 along with the results presented in 4 - is largely motivated by the work of Trevisan and Pancotti (1998); Trevisan and Uboldi (2004) and subsequent developments of the AUS (assimilation in unstable subspace) methodology. Their work focuses mainly on BLVs whereas we consider a natural extension to compute the CLV as well. Recently, another data-based algorithm has been proposed in Martin et al. (2022), using the data to reconstruct the system dynamics and then use this reconstructed dynamics to compute CLVs. Our approach differs in a fundamental way, since our basic assumption - which is also common with all the data assimilation (DA) methods - is that a dynamical model is available but we do not know the trajectory that is being observed. This naturally leads to our proposed algorithm that uses a DA algorithm to compute an optimal estimate, which is then used to compute the CLV. We note that any algorithm, including our algorithm, for computation of CLV necessarily needs to be "offline" since it requires the backward evolution from far future. But one by-product of our algorithm that is indeed "online" (without the need to future observations) is the data-based computation of BLV using only the past observations.

The second part studies the sensitivity of the LVs to noisy perturbations of a trajectory. We study how well the LVs and the associated Oseledets' subspaces spanned by these vectors are approximated when calculated by using a noisy trajectory instead of a true trajectory. We note that this is quite distinct from the question of continuity of the LVs with respect to the phase space, as has been studied in Katok and Hasselblatt (1995, sec. 19.02); Araújo et al. (2016); Dragičević and Froyland (2018); Luo and Zhao (2022). We also note that the noisy trajectories we use are neither shadowing trajectories nor are they solutions of a stochastic version of the deterministic dynamics, since the noise is added only at discrete observation times. This choice is motivated by the parallel with the data assimilated state estimates which also provide trajectories that are neither shadowing nor solutions of stochastic dynamics.

The paper is organized as follows. We begin section 2 with an overview of the mathematical definitions in section 2.1 and of an existing algorithms for computation of covariant Lyapunov vectors in section 2.2. Our proposed data-based algorithm using data assimilation for computation of CLV is presented in section 2.3, which is also used for computing the approximate CLV of perturbed trajectories, as explained in section 2.4. We briefly describe in section 2.5 the ensemble Kalman filter algorithm that we use and in section 2.6 the metrics employed for comparison of the exact and perturbed or approximate CLV and Oseledets' subspaces. The details of the models and numerical implementation are presented in section 3. The numerical results are discussed in section 4 followed by a summary of conclusions and directions of further studies in section 5.

## 2  Problem Statement

In this section, we present a brief summary of the mathematical framework for defining the Lyapunov vectors, followed by a description of the method we used for computing them (e.g. Ginelli et al., 2013; Kuptsov and Parlitz, 2012). We then discuss our proposed method for computing these vectors either (i) using state estimates obtained from the ensemble Kalman filter or (ii) using the perturbed trajectories of a dynamical system. We also discuss the metrics that we use, namely the subspace angles between Oseledets subspaces, to assess the accuracy of the approximate Lyapunov vectors obtained using these two methods -





this is accuracy with respect to the Lyapunov vectors obtained from the exact trajectory (or rather, its numerical approximation by the RK4 algorithm that we use).

## 2.1 Definition and importance of covariant Lyapunov vectors (CLV)

We consider autonomous continuous time dynamical system represented by ODE of the form

$$\dot{x}_t = f(x_t), \quad \text{where,} \ x_t \in S \subseteq \mathbb{R}^n \ \text{and} \ f : S \to \mathbb{R}^n \ \text{is the vector field.} \tag{1}$$

Associated to such an ODE, the evolution for the infinitesimal perturbations in the tangent space is obtained by linearizing along a trajectory, thus leading to the following ODE for $z_t \in \mathbb{R}^n$:

$$\dot{z}_t = J(x_t)z_t, \quad \text{where} \ J(z) \ \text{is the jacobian matrix given by} \quad J_{ij}(z) = \frac{\partial f_i(z)}{\partial z_j}. \tag{2}$$

A fundamental matrix solution of this linear non-autonomous equation solves $\dot{Q}_t = J(x_t)Q_t$ with any non-singular matrix $Q_0 \in \mathbb{R}^{n \times n}$ as an initial condition. In the discussion below, we consider the evolution of the trajectory and the corresponding perturbations at times $\ldots, t_k, t_{k+1}, \ldots$, and use the notation $x_k \equiv x_{t_k}$, $z_k \equiv z_{t_k}$, etc. With this notation, using the fundamental matrix solution, the tangent linear propagator from time $t_k$ to $t_l$ can be written as

$$\mathcal{M}_{k,l} = Q_l Q_k^{-1} \quad \text{with the property that} \quad z_l = \mathcal{M}_{k,l} z_k. \tag{3}$$

The eigenvectors and eigenvalues of $\mathcal{M}_{k,l}$ for large $l \to \infty$ and $k \to -\infty$ capture the asymptotic stability properties of the perturbations around a trajectory and will be the primary focus in this paper. The existence of these limits is the main content of the Oseledets' multiplicative ergodic theorem (Oseledec, 1968; Oseledets, 2008). Specifically, under suitable conditions, the theorem implies the existence of the following limits:

$$\lambda^+(z, t_k) := \lim_{t_l \to \infty} \frac{1}{|t_l - t_k|} \log \frac{\|\mathcal{M}_{k,l} z\|}{\|z\|}, \quad \text{and} \quad \lambda^-(z, t_l) := \lim_{t_k \to -\infty} \frac{1}{|t_l - t_k|} \log \frac{\|\mathcal{M}_{k,l} z\|}{\|z\|}, \tag{4}$$

where $\lambda^\pm(z, t_k)$ are called the Lyapunov characteristic exponents. Under appropriate regularity conditions, the two limits in (4) give the same set of Lyapunov exponents but with opposite sign and we drop the superscript $\pm$ for the exponents. There are at most $n$ distinct such exponents, which are usually ordered as $\lambda_1 > \lambda_2 > \cdots > \lambda_s$ with $s \leq n$. Oseledets' theorem also proves the existence of the Oseledets subspaces

$$\phi = S_{s+1}^+ \subsetneq S_s^+ \subsetneq \cdots \subsetneq S_1^+ = \mathbb{R}^n \tag{5}$$

with the property that $\lambda(z, t_k) = \lambda_i$ when $z \in S_i^+ \setminus S_{i+1}^+$. Such vectors $z$ satisfying this latter property are called the *forward Lyapunov vectors*. The Oseledets subspaces corresponding to the $t_k \to -\infty$ limit in (4) are given by

$$\phi = S_0^- \subsetneq S_1^- \subsetneq \cdots \subsetneq S_s^- = \mathbb{R}^n \tag{6}$$

and analogously define the *backward Lyapunov vectors*. Note that these subspaces depend on time, i.e. on $t_k$ (resp. $t_l$) for forward (resp. backward) Lyapunov vectors, so more precisely, $S_j^+ = S_j^+(t_k, x_k)$ etc. For autonomous dynamical systems, this





time dependence is only through the trajectory, i.e., on the phase space point $x_k = x(t_k)$. Thus, more precisely, $S_j^+ = S_j^+(x_k)$. Though this dependence was dropped above for simplicity of notation, exploring the dependence of the Oseledets subspaces and the Lyapunov vectors on the phase space trajectories is one of the main aims of this paper, as we discuss below.

The above discussion naturally raises the question of whether nearby points in phase space have Oseledets spaces that are
125 "close" to each other in an appropriate metric. In other words, this is a question of continuity of these Oseledets spaces with respect to phase space and has been investigated theoretically in Katok and Hasselblatt (1995); Araújo et al. (2016); Dragičević and Froyland (2018); Luo and Zhao (2022), proving Hölder continuity of these spaces, and numerical methods for computing derivatives of CLVs has been developed recently in Chandramoorthy and Wang (2021). But we are not aware of any numerical study of the continuity of the Oseledets spaces with respect to perturbations to the full trajectory. The main focus of this paper
is precisely to address this lacunae by numerically studying the sensitivity of the Lyapunov vector to perturbations in phase space. One of the key difficulties in these numerical investigations is explained in detail in section 2.2.

The forward and backward Lyapunov vectors are not mapped to each other under the action of the tangent linear operator, i.e., they are not covariant. Further they are also not invariant with respect to time reversal. Instead they satisfy the following property: if the forward (resp. backward) Lyapunov vectors at time $t_k$ are arranged in columns of $\Phi^+(t_k)$ [resp. $\Phi^-(t_k)$], then

$$\mathcal{M}_{k,l}\Phi^+(t_k) = \Phi^+(t_l)L_{k,l} \qquad \text{and} \qquad \mathcal{M}_{k,l}^{-1}\Phi^-(t_l) = \Phi^-(t_k)R_{k,l}, \tag{7}$$

where $L_{k,l}$ and $R_{k,l}$ are, respectively, lower and upper triangular matrices. The diagonal elements of these matrices give the local stretching or contraction of the Lyapunov vectors. These properties motivate the numerical algorithms to calculate the BLV and FLV, e.g., see the review Kuptsov and Parlitz (2012).

To summarize, the BLV and FLV are not covariant but form orthonormal bases of the Oseledets subspaces. On the other hand,
these subspaces are covariant under the linear dynamics as indicated by (7). By looking for bases that may not be orthonormal, it is possible to find a set of basis vectors of these Oseledets' spaces that are covariant with respect to the dynamics and invariant with respect to time reversal. Such basis vectors are called *covariant Lyapunov vetors* (CLV). In particular, they have the property that the $i$-th covariant Lyapunov vector $q_i(t_k)$ satisfies the dynamics given by,

$$q_i(t_l) = \mathcal{M}_{k,l}q_i(t_k) \qquad \text{and} \qquad \|\mathcal{M}_{k,k+l}q_i(t_k)\| \sim e^{\lambda_i t_l} \quad \text{for} \quad t_l \to \pm\infty. \tag{8}$$

The FLV satisfy the above only in the limit of $t_l \to +\infty$ but not in the other limit of $t_l \to -\infty$, and similarly the BLV satisfy only one of the two limits above.

The existence of such covariant Lyapunov vectors is guaranteed by the following fact: the dimensions of $i$-th forward and backward Oseledets subspace $S_i^+$ and $S_i^-$ are, respectively, $d_s + \cdots + d_i$ and $d_1 + \cdots + d_i$. Since the sum of these dimensions is $n + d_i$, their intersection has minimum dimension of $d_i$. We can see that the vectors that belong to this intersection $S_i^+ \cap$
$S_i^-$ satisfy the properties (8) and are the covariant Lyapunov vectors that we seek. Indeed the numerical algorithm presented below directly make use of the fact that they belong to this intersection.



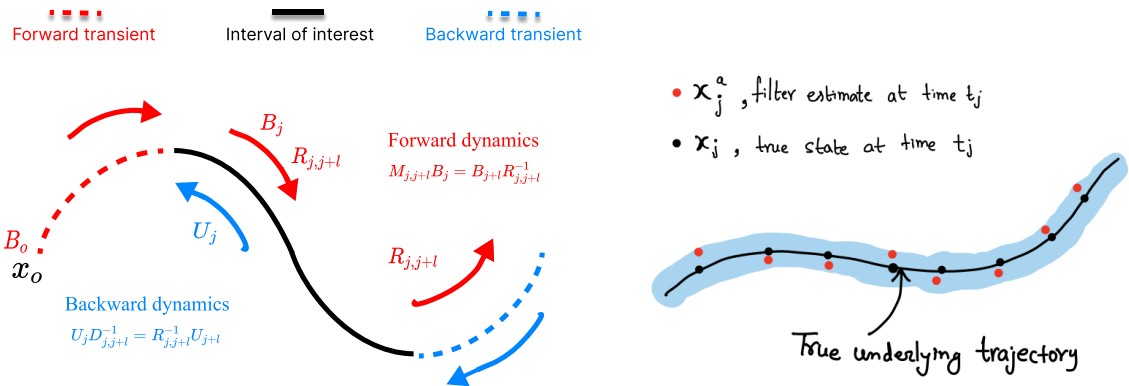

**Figure 1.** Left: schematic diagram of Ginelli's dynamic algorithm. Right: Using filter estimated trajectory as an approximate trajectory for computing Lyapunov vectors.

## 2.2 Computation of Lyapunov vectors

We now discuss the dynamic algorithm introduced in the work of Ginelli et al. for the computation of BLVs and CLVs about a reference trajectory. We assume that we have the database consisting of states $x_j$ sampled at time $t_j = j\Delta t$ with $\Delta t$ as the interval between two consecutive states from a fixed trajectory starting at $j = 0$. Since we have pre-computed trajectories, we first carry out time integration of the perturbation vectors in tangent space, using the state information from the trajectory whenever required in the evaluation of the jacobian in equation (2) to obtain the BLVs. More specifically, BLVs are required as an intermediate step since they provide a basis in the tangent space which is then used to represent the CLVs at any time $t_j$. This necessitates that the trajectory should be long enough to contain three distinct time intervals: (i) an initial "forward transient" time interval denoted as $[0, I]$ to account for the forward transient required to converge to the BLV basis, (ii) the subsequent interval of interest denoted by $[I, F]$ over which we want to obtain the BLVs (and later on the CLVs), and (iii) an additional "backward transient" interval denoted by $[F, E]$ to account for the backward iterations to converge to give the CLVs. This is shown schematically in the left panel of figure 1.

We perform gram-schmidt re-orthonormalization via QR decomposition after every $l\Delta t$ interval and use or store the vectors for integrating over the next interval. The number $N_{0I}$ of times the QR-decomposition is performed in the forward transient time interval is given by the relation $N_{0I}l\Delta t = I$. Similarly for the other two intervals, $N_{IF}$ and $N_{FE}$ denote the number of times the QR-decomposition is performed, i.e., $N_{IF}l\Delta t = F - I$ and $N_{FE}l\Delta t = E - F$.

Let $B_j \in R^{n \times m}$ with $m \le n$ denote the matrix containing a set of orthogonal perturbation vectors along the columns at time $t_j$. For the forward transient interval, we initialize $B_0$ as the initial condition for equation (2) and integrate over small time intervals $[t_j, t_{j+l}]$ of length $l\Delta t$, at the end of which we perform the gram-schmidt re-orthonormalization of the columns of $B_j$. Recall that $\mathcal{M}_{j,j+l}$ is the linear tangent propagator from time $t_j$ to $t_{j+l}$. So the evolution of the perturbation vectors followed



by re-orthonormalization satisfies the following relations,

$$\tilde{B}_{j+l} = \mathcal{M}_{j,j+l} B_j \quad \text{(evolution)} \quad \text{and} \quad \tilde{B}_{j+l} = B_{j+l} R_{j,j+l} \quad \text{(re-orthonormalization)}, \tag{9}$$

for $j = 0, l, \ldots, (N_{0I} - 1)l$ where, $R_{j,j+l}$ is the matrix containing the growth rates of the vectors over the interval $[t_j, t_{j+l}]$ obtained by QR decomposition of $\tilde{B}_{j+l}$. The choice of $l$ must be small enough so as to prevent the rank collapse of the columns of $\tilde{B}_j$ at any time $t_j$ where the matrices are stored. The length of the forward transient interval $[0, I]$ is chosen to be long enough such that at time $I$, the columns of the matrix $B_j$ provide a good approximation of the columns of $\Phi_j^-$, i.e., the BLVs, as defined in equation (2).

From this time onwards, we store both the matrices $B_j$ and $R_{j-l,j}$, as computed in (9) for $j = (N_{0I})l, \ldots, (N_{0I} + N_{IF} - 1)l$, thus covering the time interval $[I, F]$ during which we want to compute the CLVs. Note that over this time interval, we already get the numerically approximate BLVs as the columns of the matrices $B_j$. The main idea is to note the following fact: if the CLVs at time $t_j$ are arranged as columns of matrix $C_j$, then their relation to the BLV basis can be expressed as

$$C_j = B_j U_j, \tag{10}$$

for some $U_j$ which is an upper triangular coefficient matrix. This is due to the fact that the $i^{th}$ CLV lies in the span of the first $i$ BLVs. Since the CLV are covariant, they satisfy the following relation:

$$C_{j+l} D_{j,j+l} = \mathcal{M}_{j,j+l} C_j, \tag{11}$$

where $D_{j,j+l}$ is a diagonal matrix with the diagonal elements being the norms of the columns of the product on the right-hand side. Combining the above three relations (9)-(11), the backward evolution of the perturbation vectors in matrices $U_j$ followed by re-normalization satisfies the following relations,

$$\tilde{U}_j = R_{j,j+l}^{-1} U_{j+l} \quad \text{(backward evolution)} \quad \text{and} \quad \tilde{U}_j = U_j D_{j,j+l}^{-1} \quad \text{(re-normalization)}. \tag{12}$$

Hence the rest of the algorithm is the backward evolution aimed at calculating these matrices $U_j$.

At the end of the interval $[I, F]$, we further integrate forward the perturbation vectors using (9) for $j = (N_{0I} + N_{IF})l, \ldots, (N_{0I} + N_{IF} + N_{FE} - 1)l$, thus covering the time interval $[F, E]$. During this time, we only store the matrices $R_{j-l,j}$. At the end of this backward transient interval, we set $U_j$ for $j = (N_{0I} + N_{IF} + N_{FE})l$ to be a generic upper triangular full-rank matrix $U_F$. This is to ensure that the $i^{th}$ column of the matrix $C_j$ lies in $i^{th}$ backward Oseledets subspace $S_i^-$ at $j = (N_{0I} + N_{IF} + N_{FE})l$, since under the backward evolution, a generic vector in this subspace $S_i^-$ converges asymptotically to the $i^{th}$ CLV.

The backward dynamics is then performed on the upper-triangular coefficient matrices $U_j$ over the backward transient interval via (12) for $N_{FE} + N_{IF}$ number of steps for $j = (N_{0I} + N_{IF} + N_{FE} - 1)l, \ldots, (N_{0I} + N_{IF})l, \ldots, (N_{IF})l$, utilizing the inverses of $R_{j-l,j}$ which were computed and stored in the course of forward evolution over the respective time interval. This step gives us the set of upper triangular matrices $U_j$ over the time interval $[F, E]$ which is the backward transient interval and also over the interval $[I, F]$ over which the relation (10) between BLVs and CLVs is valid with the matrices $U_j$. Thus over the interval $[I, F]$, we obtain the BLVs that are stored in the matrices $B_j$, and using the upper triangular coefficient matrices $U_j$, we also obtain the CLVs as columns of $C_j = B_j U_j$.



For an more in-depth discussion of the algorithm and its convergence, we refer to Ginelli et al. (2013); Kuptsov and Parlitz
(2012) and Noethen (2019).

## 2.3    Data-based algorithm to calculate the Lyapunov vectors

We now describe the problem of computing the LVs when we cannot observe the full system in time i.e. we only have access
to partial and noisy observations of some of the components of the state $x_j$ instead of the full trajectory. As discussed above in
section 2.2, Ginelli's algorithm requires a reference trajectory or the initial condition at $t_j = 0$ which can be integrated forward
to obtain that reference trajectory. The above procedure cannot be carried out directly when the initial condition is unknown
and only partial observations of the state over time are available. This is because of the exponential divergence of nearby
trajectories for chaotic systems.

Under the condition that the true trajectory may only be observed partially and indirectly through noisy measurements of
state-dependent physical quantities, nonlinear filtering aims to obtain optimal estimates of the state that are in proximity to
215 the true trajectory, or more precisely, the posterior probability distribution called the filtering distribution or simply the filter.
(See, e.g., Carrassi et al., 2018; Asch et al., 2016; Fletcher, 2017, for reviews and further references.) Most common numerical
filtering algorithms compute Monte Carlo approximations of the filtering distributions, and the mean of the filter – called the
analysis mean – is an optimal estimate of the true state.

When a numerical filter performs reasonably well, we expect the analysis mean to be sufficiently near the true state. The
220 filter may also be used to give an uncertainty associated with the analysis mean, most commonly in terms of the covariance of
the filter. One of the important factors affecting this uncertainty is the observational uncertainty and we focus on this aspect
in this paper. Some of the other factors affecting the filter performance include observational frequency, the sparsity of the
observations, and the dynamical characteristics of the system itself.

We propose to use the analysis mean $x_j^a$ obtained over time as an approximation of the state $x_j$ of the true underlying
trajectory, in order to compute the approximations of the true Lyapunov vectors and Oseledets' subspaces. This leads to a our
proposed modification of Ginelli's algorithm discussed in section 2.2: the (pseudo-)trajectory we use in this algorithm now
consists of the analysis means $\{x_j^a\}$ at times $t_j$ obtained from a filtering algorithm. In order to get the tangent linear propagator
$\mathcal{M}_{j,j+1}$ over the time interval $(t_j, t_{j+1})$, equations (1)-(2) are solved with $x_j^a$ as the initial condition for (1) and with $B_j$ as the
initial condition for (2). The other steps are exactly the same as described in the previous section 2.2.

This allows us to compute the LVs and the sub-spaces defined by them from the estimated trajectory obtained from any
general data assimilation method. However, this comes with a caveat that nonlinear filtering algorithms result in an estimated
trajectory that is not a dynamical trajectory of the model itself i.e. there is no initial condition such that if integrated forward
in time contains the filter analysis means obtained over time. But it is close to the true state over time which can be quantified
by the $l_2$ error over time or other popular metrics such as RMSE. In this paper, we use ensemble kalman filter (EnKF) as our
choice of filtering algorithm to obtain the analysis mean trajectory. We apply this method to two models L63 and L96. Further
details of EnKF and the data assimilation experimental setup used in this paper are given in sections 2.5 and 3, respectively.





## 2.4 Lyapunov Vectors from perturbed trajectory

Any filter-based trajectory violates the criteria of being the dynamical trajectory of the system. Even when the deviations $e_j^a$ of the analysis from the true state, given by $e_j^a = x_j^a - x_j$ are small, it is not *a priori* clear how they may affect errors in the computed LVs, through their effect on the Jacobian matrix in equation (2). This motivates us to investigate the stability of the LVs from a more general perspective.

To systematically investigate the stability of numerically calculated LVs about the reference trajectory, we generate a perturbed or noisy version of the state $x_j$ at each point on the trajectory by adding random perturbation to the true underlying trajectory in the following way,

$$\tilde{x}_j = x_j + \epsilon_j, \quad \text{with} \quad \epsilon_j \sim \mathcal{N}(0, \sigma^2 \times I_d), \tag{13}$$

where $\epsilon_j$ is randomly sampled from a standard normal distribution of covariance $\sigma^2 \mathbf{I}$ at time $t_j$ and $\tilde{x}_j$ is then used as state estimates in the computation of the BLVs and CLVs at the respective time. We refer to the obtained trajectory as a perturbed orbit for the respective noise level $\sigma$ in future discussions. We show the results for $\sigma \in \{0.1, ..., 0.5, 1.0, ...5.0\}$ for the Lorenz-63 model and for $\sigma \in \{0.1, ..., 0.5\}$ for the Lorenz-96 model.

We compute all the BLVs and CLVs about the true and the perturbed trajectories over a common interval using the same procedure mentioned in section 2.2. The above notion of sensitivity to the perturbations in the underlying dynamical trajectory allows us to think of the analysis mean trajectory as a perturbed trajectory obtained from the true trajectory where the perturbations follow the unknown error statistics of the difference between the true state and analysis mean over time. We perform sensitivity analysis for L63 and L96 for $d = 10, 20$ and $40$, which we describe in section 3.1. As far as we know, this question of the sensitivity of BLVs and CLVs to perturbations in the space of trajectories has not been investigated either numerically or mathematically. We note that the results about their Hölder continuity with respect to initial conditions (Dragičević and Froyland, 2018; Luo and Zhao, 2022; Araújo et al., 2016) are quite distinct from the question of continuity with respect to perturbations of the whole trajectory.

## 2.5 Ensemble Kalman Filters

Introduced in the context of data assimilation in Evensen (2003), ensemble kalman filters are a Monte-Carlo approach of approximation of the original Kalman filter which is a sequential state estimation algorithm (Cohn, 1997) based on the assumption of linearity and gaussianity of the probability distributions involved to recursively compute from the prior distribution, the posterior distribution incorporating the latest observation by simply updating mean and covariance. The filter employs an ensemble representation of probability distributions which is a collection of states sampled from the respective pdfs. This approach is particularly useful when the dynamics is nonlinear as each ensemble member can be integrated individually. In between the observations, each member of the ensemble is evolved in time according to the model equations. When a new observation arrives, the mean and covariance required in the Bayesian update is replaced by the empirical mean and covariance computed from the ensemble respectively. Localization and inflation are ad-hoc methods that make EnKF work with small ensemble



sizes, making it suitable for large dimensional systems and a practical choice for operational data assimilation (Carrassi et al.,
2018). We use EnKF for performing the twin experiments with L63 and L96 models using noisy and partial observations to
perform assimilation.

## 2.6   Comparison metrics

We compare the individual LVs and the Oseledets subspaces obtained using the perturbed or the assimilated trajectories with
those obtained from the true underlying trajectory. To understand the sensitivity of the individual LVs with respect to the
perturbation strength $\sigma$, we use the angle between the $i^{th}$ Lyapunov vector computed from the original trajectory and the
perturbed trajectory. Note that we compute the Lyapunov vectors over a length of trajectory (of course discretely sampled) and
at each of the phase space points of that trajectory, we compute the cosine of the angle between the LVs of the true trajectory
and LVs of the approximate trajectory. We then plot the median of the angle computed over the sampling interval along with
the error bars which represent the $25^{th}$ and $75^{th}$ percentile of the angles, see, e.g., figures 2-3.
A subspace of dimension $k$ in $R^n$ admits an infinite number of possible basis vectors. Thus, even though the individual
LVs from the perturbed trajectory and those from the true trajectory may differ, the Oseledets subspaces spanned by them may
still be "similar." In order to quantify this "similarity" and understand the sensitivity of the Oseledets subspaces, we use the
principal angles as described in detail below.

Principal angles (Jiang, 1996) are defined as a sequence of minimum angles between two unit vectors corresponding to each
of the two subspaces, such that the unit vectors are chosen to minimize the angle between them while being orthogonal to
all the previous unit vectors obtained in their respective subspaces. Mathematically, for two subspaces $\mathcal{P}$ and $\mathcal{Q}$, the principal
angles are defined by an m-tuple of angles $\theta(\mathcal{P}, \mathcal{Q}) = [\theta_1, \theta_2, .., \theta_m]$, where $m = \min(\dim(\mathcal{P}), \dim(\mathcal{Q}))$ and $\theta_k$ is defined by,

$$cos(\theta_k) = \max_{p_k \in \mathcal{P}} \max_{q_k \in \mathcal{Q}} |p_k^T y_k| \quad \text{subject to } ||p_k|| = ||q_k|| = 1, \quad p_k^T p_i = q_k^T q_i = 0, \quad i < k. \tag{14}$$

When the set of angles between two subspaces are all small, they are closely aligned and have a strong degree of similarity.
On the other hand, if the angles are large, this implies that the subspaces are more dissimilar and have less overlap. The two
subspaces are identical (when they have the same dimension) or one is fully contained in the other (when they have different
dimensions) if and only if all the principal angles are zero.

In order to investigate the quality of subspaces we obtain from the approximate LVs, we study $i \leq k$ principal angles between
the subspace spanned by the first $k$ BLVs computed from the true and the perturbed trajectory. The aim is to understand whether
a set of $k$-BLVs have a common lower dimensional subspace. These results are discussed in section 4.4. Similar to the relative
angle of BLVs and CLVs, we compute the principal angles for different points along the trajectory and plot the median along
with $25^{th}$ and $75^{th}$ percentile for the confidence intervals, see, e.g., figure 4).

There are several methods for calculating the principal angles between two subspaces, including the singular value decom-
position (SVD) and the QR decomposition. If the two orthogonal bases $p_i$ and $q_i$ are arranged along the columns of two
matrices $P$ and $Q$ respectively, we let $P^T Q = U \Sigma V^T$ denote the singular value decomposition of $P^T Q$. Then the cosines of
the principal angles are the diagonal elements of the matrix $\Sigma$.



## 3  Experimental setting

### 3.1  Models

Since their first introduction by Ed Lorenz who used them to understand the predictability of the atmospheric convection
process, Lorenz-63 (Lorenz (1963)) and Lorenz-96 (Lorenz (1995)) models have been extensively studied, and have contributed
significantly to understanding and development of nonlinear dynamics. Lorenz-63 model was one of the first low-dimensional
models where the emergence of chaos in low dimensions was studied. We use the standard Lorenz-63 model

$$\frac{dx}{dt} = \sigma\left(y - x\right), \quad \frac{dy}{dt} = x\left(\rho - z\right) - y, \quad \frac{dz}{dt} = xy - \beta z, \tag{15}$$

with the parameters $(\sigma, \rho, \beta) = (10, 28, 8/3)$ for which the system exhibits chaos and has the well-known butterfly-shaped
attractor. The process where observations and the model dynamics are "combined" to estimate the model state or its prob-
ability distribution that is close to the true state under some metric is broadly called data assimilation (DA) Lorenz-96 is a
$n$-dimensional nonlinear, dissipative model with a constant external forcing term. It mimics the dynamics of a meteorological
scalar variable along the latitude. The model is given by the evolution of a set of $n$ ordinary differential equations given below:

$$\frac{dX_k}{dt} = X_{k-1}\left(X_{k-2} - X_{k+1}\right) - X_k + F \tag{16}$$

where $X_k$ is the $k^{th}$ component of the n-dimensional state and with periodic boundary conditions $X_{k+n} = X_k$. For increasing
values of $F$, the behavior of the system changes from being stable to weakly and then strongly chaotic. Its extensive chaotic
properties for different regimes of forcing and different dimension has been studied in Karimi and Paul (2010). For the specific
value of forcing $F = 8$ for $n = 40$ dimensions, it is a chaotic system with 13 positive Lyapunov exponents and has Kalpan-
Yorke dimension close to $28.4$. Due to its rich dynamical properties, it has served as a computationally tractable model for
performing and evaluating data assimilation twin experiments used to benchmark the performance of many data assimilation
algorithms (see, e.g., Carrassi et al. (2018)) before being applied to very large-scale atmospheric models.

### 3.2  Details of computing Lyapunov Vectors from filtered and perturbed trajectories

For the Lorenz-63 model in equation (15), we start with a random initial condition and integrate for a long transient up to $t = 500$ to reach the attractor. We then choose this point on the attractor as the initial condition for the true orbit which is generated
by numerically integrating the ODE for a total time $T = 350$ with $\delta t = 0.002$ using Runge-Kutta $4^{th}$ order scheme and storing
the state every $\Delta t = 0.01$. To generate observation for the data assimilation experiment, we choose to observe only the $Y$-
coordinate at every $\Delta t = 0.01$ with noisy observations given by $y_j = H x_j + \eta_j$ with $H = [1, 0, 1]$ and $\eta_j \sim \mathcal{N}(0, \mu^2)$. We show
the results for $\mu = 0.1, 0.3, 0.5, 0.7, 0.9$. To start the assimilation, we use an arbitrary initial distribution $\mathcal{N}(x_0 + 6 \times 1_3, 2.0 \times I_3)$
at $t_{j=0}$, from which we generate $N = 25$ ensemble members to initialize the EnKF algorithm. We then assimilate the previously
generated observations $y_j$ performing 35000 assimilation steps. We didn't use any inflation and localization for this simple case
of Lorenz-63 ODE. Neglecting the first 5000 assimilation steps, we perform the computation of both BLV and CLV over a
smaller interval excluding the transient intervals as mentioned below.





For the computation of LVs using the algorithm explained in section 2.2, the forward transient $[0, I]$, the sampling interval of interest $[I, F]$, and the backward transient $[F, E]$ are all chosen to be equal to $100$ which was found to be sufficient for the

convergence to BLVs. The QR decomposition is performed every $l = 1$ step. For the choice of initial perturbations $B_0$, we use the standard orthonormal basis vectors and integrate them forward in time using the tangent linear equations described in section 2.4.

For the case of the Lorenz-96 model, we perform data assimilation for $40$-dimensional model, where we assimilate $20$ observations taken at the evenly indexed components of the full state every time step of $\Delta t = 0.05$. We then generate the observations

using $y_j = H x_j + \eta_j$ with $H$ being the appropriate matrix to observe the alternate coordinates and $\eta_j \sim \mathcal{N}(0, \mu^2 I_{20})$ for the observation noise statistics. The initial condition for the filter was chosen to be the distribution $\mathcal{N}(x_0 + 5.0 \times 1_3, 1.0 \times I_{40})$ which is biased, as may happen often in practice. We implement covariance localization using the localization radius $r = 4$ when performing the analysis as it is necessary when we have partial observations with ensemble sizes smaller than the system dimensions (Carrassi et al., 2018). We vary $\mu$ as a parameter to study the dependence on observation noise in the reconstruction

of the trajectory by relating it to the $RMSE$ of the obtained analysis trajectory.

To study the dimension dependence of sensitivity in L96, we perform similar numerical experiments in dimensions equal to $10, 20$ and $40$. We generate a long trajectory of length $T$, where for dim. $10$ and $20$, we choose $T = 600$ whereas for $40$ we choose $T = 1000$, all obtained using the solver time step of $\delta t = 0.01$ while saving the states every $\Delta t = 0.05$ s. For dimensions $10$ and $20$, we chose the forward transient interval $[0, I]$ and backward one $[F, E]$ to be of length $200$ whereas for dimension

$40$, we choose them to be $400$, and the interval of interest $[I, F]$ is of length $200$. The QR decomposition is performed every $l = 5$ steps. The lengths of forward and backward transients were chosen based on the convergence of BLVs for the respective systems.

## 4   Results

We now discuss the results of using the algorithms described above, by comparing the true LVs and the LVs obtained from

the assimilated and the perturbed trajectories for both Lorenz-63 and Lorenz-96 systems. In section 4.1, we first describe the results obtained from using assimilated trajectory for different observation noise strength $\mu$. We plot the median of the angle between the true vectors and the ones obtained from the assimilated trajectory along with a confidence interval denoting the $25^{th}$ and $75^{th}$ percentile, obtained from a sample of points at which we compute the LV in the time interval $[I, F]$ along the trajectory. In section 4.2 and 4.3, we present the results of the exploration of the sensitivity of the BLVs and the CLVs to the

noise strength of the perturbations in the underlying true trajectory.

For Lorenz-96 in 40 dimensions, we discuss, in section 4.4, the approximations of the Oseledets subspaces obtained using the assimilated and the perturbed trajectories by plotting the principal angles between $k$-dimensional subspaces spanned by the first $k$ BLVs obtained from the true and the approximate trajectory for k = 2, 5, 10, 15, and 20. To further understand the quality of recovered subspaces from the approximate trajectories, we compare them against principal angles between randomly




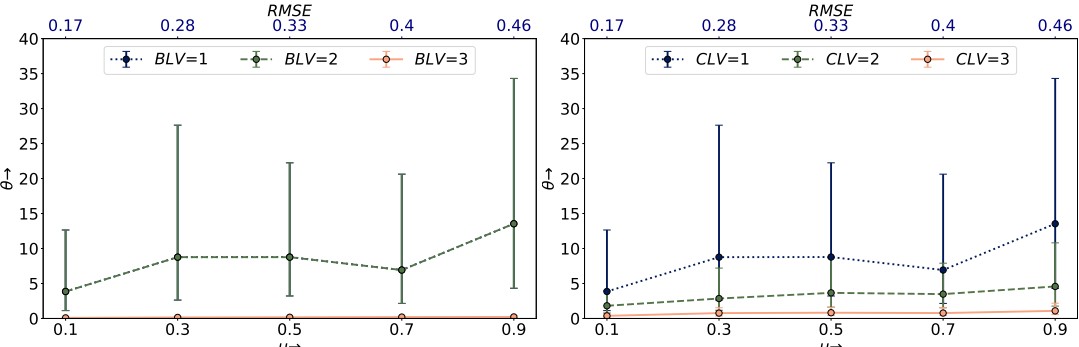

**Figure 2.** The figure shows the angle $\theta$ between the true LVs and those recovered from the analysis trajectory, for the BLVs (left) and the CLVs (right) for the Lorenz-63 model, for different levels of observational noise $\mu$ (bottom axis) along with the corresponding RMSE (top axis) of the analysis trajectory. The dots represent the median and the error bars represent the $25^{th}$ and $75^{th}$ percentiles.

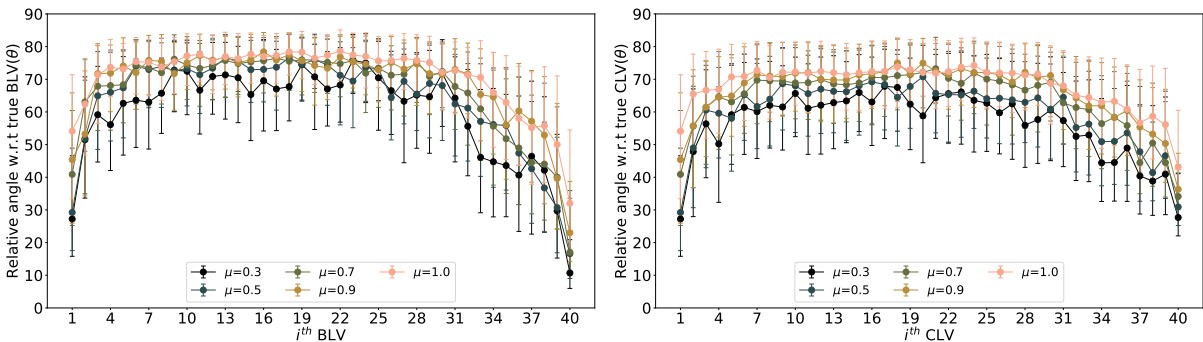

**Figure 3.** The figure shows the angle between the true LVs and those recovered from the analysis trajectory, for the BLVs (left) and the CLVs (right) for the 40-dimensional Lorenz-94 model. Different lines are for different observational noise levels $\mu$. The dots represent the median and the error bars represent the $25^{th}$ and $75^{th}$ percentiles.

generated $k$-dimensional subspaces, noting that the angles for randomly generated subspaces are significantly larger than those between the true and perturbed Oseledets' subspaces.

## 4.1   BLVs and CLVs computed from assimilated trajectories

Figure 2 shows the angles between the Lyapunov vectors obtained from the true trajectory with those from the assimilated trajectory for Lorenz-63 model. In this case, we only observe the $y$-coordinate. Left panel shows the angles for the BLV while
the right one is for the CLV. We first note that since BLV are orthonormal, two of these angles are necessarily equal which happen to be those between the second and third BLV and they are quite small even for the largest observational noise strength we have used. In addition we note that the median of angle between the first - most unstable - BLV is also within 15 degrees and does not increase rapidly with the observation noise strength $\mu$. The CLVs show a similar behaviour but being non-orthonormal





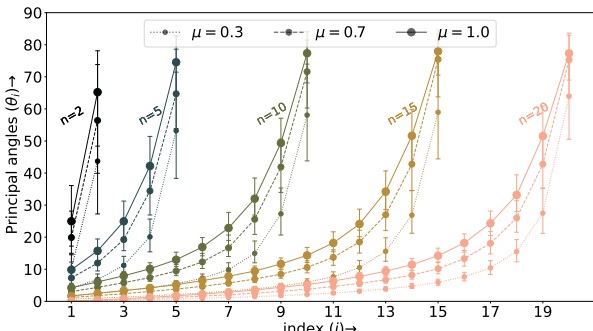

**Figure 4.** The set of principal angles between $n$-dimensional Oseledets' subspace and the corresponding subspace recovered from the analysis trajectory for different values of observational noise level $\mu = 0.3, 0.7, 1.0$ for $n = 2, 5, 10, 15, 20$ for Lorenz-96 in 40 dimensions.

basis, all the three angles are distinct, with the angle between the third - most stable - CLV being the smallest. The top axis in these plots shows the RMSE averaged over the whole time interval of interest $[I, F]$ of length 100 in this case. We later discuss the relation of these results to the case of perturbed trajectory discussed in detail in section 4.2.

For Lorenz-96 in 40 dimensions, we performed assimilation by observing 20 alternate coordinates. We compute all the 40 BLVs and CLVs from the assimilated trajectories for different observation noise strengths $\mu$ and the LV index versus the error in the angle. We see that apart from the first few most unstable and the last few most stable LVs, the angles between the true and approximate LVs are quite large, being greater than $45°$. Even the angles for the first - most unstable - LVs are larger than $20°$. This indicates that the LVs obtained from the assimilated provide a very poor approximation of the true LVs. This behaviour is quite distinct from the low-dimensional Lorenz-63 model for which the assimilated trajectory could be used to obtain a significantly better approximation of the true LVs, as discussed above.

Even though the individual LVs are not approximated well, it may be possible that the subspaces spanned by these vectors may have significant overlap, which is exactly the question of approximation of Oseledets' subspaces. We now investigate this question, using principle angles (PA) between these subspaces.

In figure 4, we plot the $n$ principal angles between $n$-dimensional Oseledets' subspaces obtained from the assimilated trajectory and the true Oseledet's subspaces, for $n \in \{2, 5, 10, 15, 20\}$, for three different values of observational noise. We see that with an increase in the subspace dimension $n$, the number of principal angles which are smaller than a certain threshold, say $20°$, increase almost linearly with $n$. For example, for $n = 15$ and $\mu = 1.0$, there are around 11 angles less than $20°$. his indicates within the 15-dimensional Oseledets' subspace defined by the first 15 approximate LVs, there is a 11-dimensional subspace (not necessarily Oseledets' space) which is within $20°$ of the true 11-dimensional subspace. With the increase in $\mu$, the relative angle increase systematically for all the P.A. The increase is more prominent for a higher P.A. index.

To summarize, recovering individual vectors from assimilated trajectories is not possible except for the first few and the last few LVs. But embedded within any high dimensional Oseledets' subspace, there are lower dimensional subspaces which





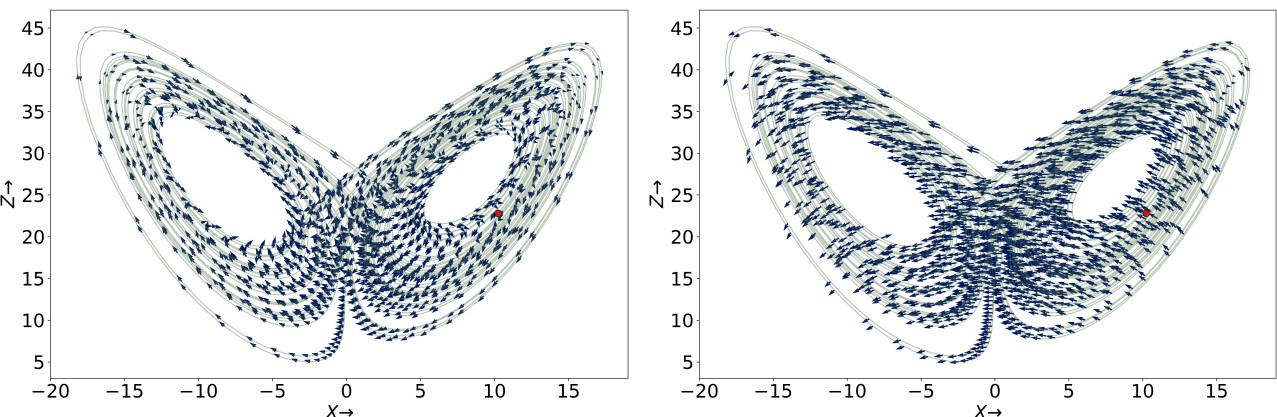

**Figure 5.** The first and the last component of the $1^{st}$ and $3^{rd}$ CLV plotted in XZ coordinates along the trajectory for Lorenz-63.

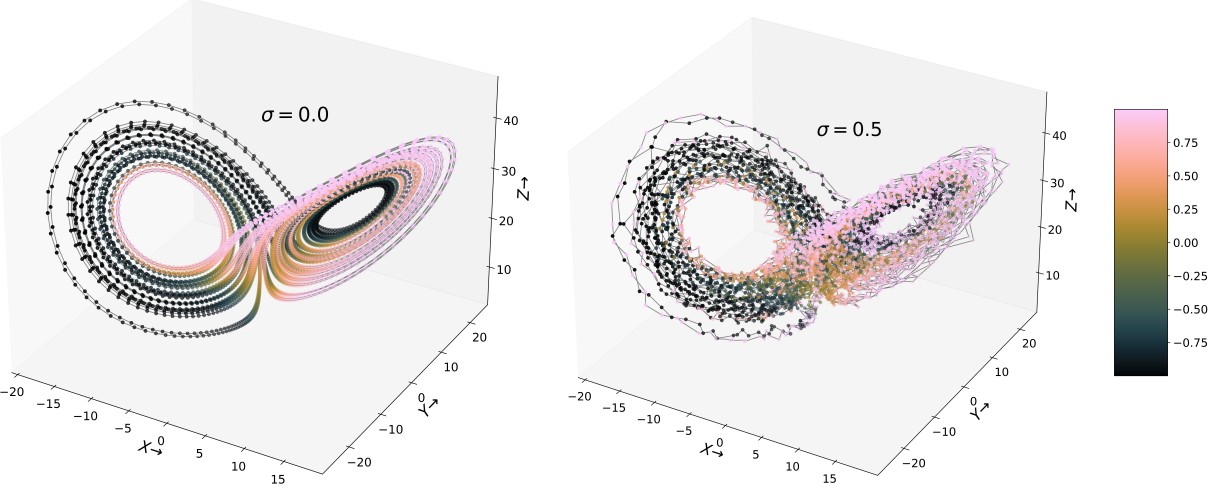

**Figure 6.** Attractor of Lorenz-63 with color indicating the cosine of the angle between the $1^{st}$ and the $2^{nd}$ CLV for unperturbed (left) and perturbed trajectory for $\sigma = 0.5$ (right).

are close to the true subspaces. In order to understand this behaviour more clearly, we now study the dependence of this approximation on the strength of perturbation of the trajectories.

## 4.2 Dependence on perturbation strength for Lorenz-63

We first show in figure 5 the geometrical structure of the $1^{st}$ and $3^{rd}$ CLV by plotting the first and the last component of the respective vectors in the XZ-plane. We observe that the orientation of the vectors seems to change continuously as one



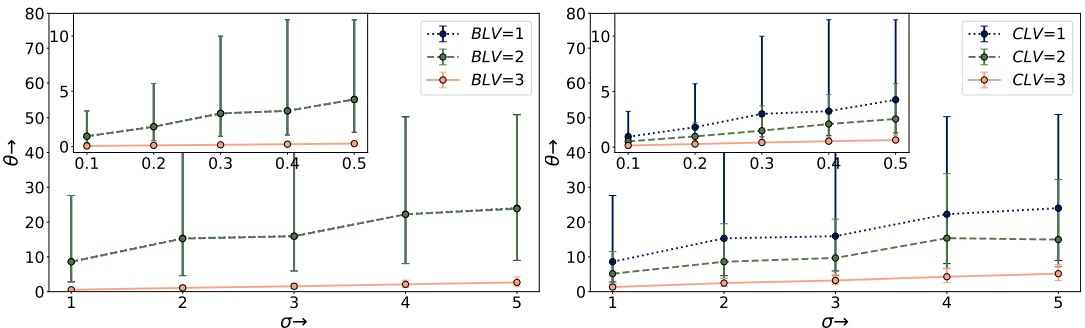

**Figure 7.** The angle $\theta$ between the true and LV from the perturbed trajectory for different perturbation strength $\sigma$ for Lorenz-63. The left and right panels show the results for the BLVs and the CLVs respectively. The dots represent the median and the error bars represent the $25^{th}$ and $75^{th}$ percentiles and the different line types for different LVs.

moves along the trajectory. In general, the mutual angle between any two CLVs change along the trajectory in the phase space. It was found recently in Brugnago et al. (2020) that these geometrical features such as the mutual angles between CLVs contain additional information, unlike FLVs and BLVs which are orthogonal basis vectors of the tangent space. Whenever the trajectory jumps from the right wing of the attractor to the left and vice-versa, the first two CLVs become parallel and

405 anti-parallel respectively. In figure 6, we use colors to plot the cosine of the angle between the first two CLVs over the attractor. We also plot the same for the perturbed trajectory for perturbation strength $\sigma = 0.5$, which has no point on the attractor with probability 1, but the angle between the first two CLVs computed from the perturbed trajectory still captures this geometrical information, although we get a fuzzy picture of attractor using the perturbed trajectory as we increase the perturbation strength $\sigma$.

To study the dependence on perturbation strength $\sigma$, we now plot in figure 7 the angle between the LVs about the true trajectory and the perturbed trajectory as a function of the noise strength $\sigma$, for the three Lyapunov vectors. The relative angle between the true and the perturbed BLV and CLV increases gradually as we increase $\sigma$ with a constant slope. The $1^{st}$ and $2^{nd}$ BLV have the same rate, whereas the rates are different for the 1st and 2nd CLVs. We also plot the absolute error in the exponents computed from the perturbed trajectory and the original trajectory. The errors in the exponents obtained are of the

order $0.2$, this tells us that they are almost unaffected by the perturbation when computed from perturbed trajectories.

### 4.3   Dependence on dimension and perturbation strength for Lorenz-96

We now describe the results for the sensitivity of the LVs for Lorenz-96. In figure 8, we plot the relative acute angle between the LVs from the true and perturbed trajectory for both BLVs and CLVs for Lorenz-96 for dimensions $d = 10, 20$, and $40$. Similar to results in the previous section 4.1 about LVs from assimilated trajectories, we observe that even with a small amount

of noise strength, the individual vectors quickly misalign from the true vectors, which is quite different compared to Lorenz-63. Only the first few most unstable vectors and the few most stable vectors corresponding to the two opposite ends of the Lyapunov spectrum have significant projection along the true vectors. For the intermediate BLVs the angles approximately



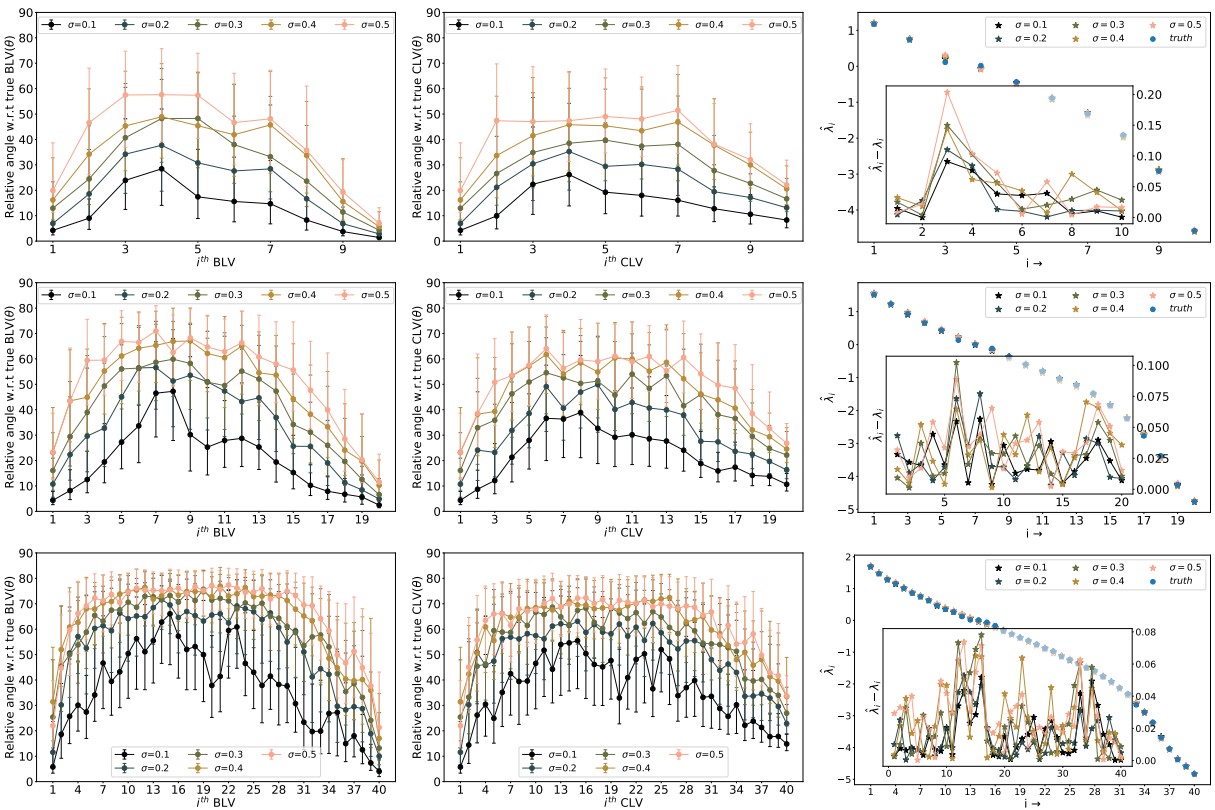

**Figure 8.** Angle $\theta$ between the individual true and recovered BLVs (left) and CLVs (middle) for different perturbation strength $\sigma$ for Lorenz-96 in $10, 20$ and $40$ dimensions, in the top, middle, and bottom rows, respectively. The third column shows the Lyapunov exponents computed from perturbed trajectories and the inset shows relative absolute errors from the exponents of the unperturbed trajectory.

lie in the interval $[45°, 90°]$. The CLVs follow a similar picture as the BLVs for the few most unstable directions and for the stable directions. The CLVs for $d = 20$ and $10$ seem to have smaller errors than their BLV counterparts. The error in angle also

increases systematically with increasing perturbation strength $\sigma$. The effect of dimension for this extensively chaotic system is clearly evident by the degrading sensitivity of both the BLVs and CLVs we double the dimension from $d = 10$ to $20$ and $40$.

In the third column in figure 8, we also plot the exponents for the true and the perturbed trajectories for different values of observational noise strength $\sigma$. The inset shows the absolute errors in the exponents obtained from the perturbed trajectories from the true exponents. We notice that these absolute errors are small of the order of $0.1$ for $d = 20$ and $40$ and $0.2$ for $d = 10$

which suggests that the exponents themselves are not as sensitive as the BLVs and the CLVs. The relative absolute errors in the exponents do not seem to follow any trend for different values of $\sigma$. This shows that the Lyapunov spectrum is quite robust to the perturbations in the underlying trajectory.





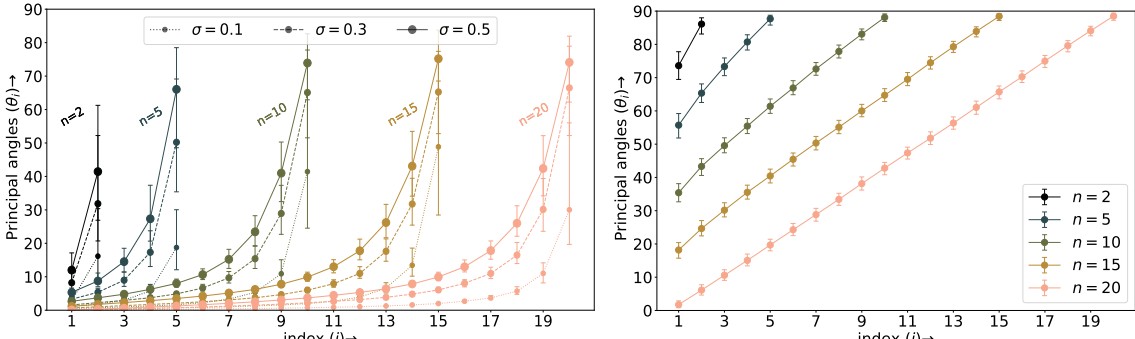

**Figure 9.** The left and the right panel shows principal angles (PA) between the $n$-dimensional Oseledet's subspace and the corresponding subspace recovered from perturbed trajectory for $k = 2, 5, 10, 15$ and 20. The line styles denote different $\sigma$ values used, the dots represent the median of the $i^{th}$ PA computed over the number of points in the sampling interval with the bars are $25^{th}$ and $75^{th}$ percentile. For comparison, the right panel shows PA between a $n$-dimensional Oseledet's subspace and a random subspace of the same dimension. The dots in the right panel represent the median with the $25^{th}$ and $75^{th}$ percentile over 100 realizations of the random subspaces as the error bars.

### 4.4 Oseledets' subspaces spanned by the LVs for different perturbation strengths

We now move towards understanding Oseledets' subspaces recovered from the perturbed trajectories instead of the individual vectors by computing the principal angles between the respective subspaces obtained from the true and the perturbed trajectories. Figure 9 shows the PA for a few $n$-dimensional subspaces for $n = 2, 5, 10, 15, 20$ over the sampling interval $[I, F]$ for $d = 40$. The behaviour is very similar to the case of using assimilated trajectory that was discussed in section 4.1 in figure 4. In particular, there are a majority of small principal angles within $30^{\deg}$ for $n = 5$ onwards, suggesting that the subspaces computed from the perturbed trajectory have significant overlap with the subspaces spanned by the true BLVs. As earlier, the angles increase with increasing perturbation strength $\sigma$.

The merits of studying the principal angles are revealed in the fact that when the angle between the individual BLVs from the perturbed trajectory is quite different, the unstable subspaces computed from them do have some similarity with the subspace spanned by the unstable vectors. When using subspaces instead of actual vectors, the BLVs from perturbed or approximate trajectories might still present some merit in capturing the unstable or stable Oseledets' subspaces. To emphasize this point further, we also plot the principal angle between a $n$-dimensional random subspace and the Oseledet's subspace from the ture trajectory. We see the number of obtained P.A. which are smaller than $30°$ are very small compared to the ones obtained from the assimilated and perturbed trajectories and these angles increase linearly with index which is quite distinct from the case of the approximate Oseledets' subspaces.




## 5 Conclusions

This paper focuses on two key questions pertaining to the computation of Lyapunov vectors when complete knowledge of the true underlying trajectory for a dynamical system is not available. Lyapunov vectors are state-dependent and their numerical computations using the commonly used algorithms by Wolfe and Samelson (2007); Ginelli et al. (2013) require the underlying trajectory or the initial condition along with the model for the system from which the trajectory can be generated. These algorithms rely crucially on a long trajectory for the convergence to the Lyapunov vectors. This poses a major challenge for

chaotic systems since even a small error in initial conditions leads to a totally different trajectory.

The first question that we address is how to use partial and noisy observations in order to compute an approximation of the Lyapunov vectors. We propose a methodology for this purpose, combining the algorithm of Ginelli et al. (2013) with the EnKF. Specifically, using EnKF for filtering, we use Ginelli's algorithm Ginelli et al. (2013) with the filter mean as an approximate pseudo-trajectory and the model linearization between the assimilation times. We demonstrate the efficacy of the proposed

idea in the context of low dimensional systems by applying it to Lorenz-63 model using only y-coordinate observations. On the other hand, for high-dimensional chaotic ODE like Lorenz-96 in various dimensions above 10, the results show that apart from a few most stable and most unstable directions, the Lyapunov vectors are very sensitive and cannot be approximated well using the filter mean as an approximate trajectory.

In order to understand the errors and / or biases in the recovered Lyapunov vectors, we naturally investigate the second

question: how does the perturbation strength affect the LVs obtained from approximate trajectories. In particular, we explore the sensitivity of the numerical computation of both the BLVs and the CLVs to general perturbations in the underlying trajectory. In small dimensions, the results for Lorenz-63 show that the recovered vectors are quite close to the true ones even for significant perturbation strength. This naturally explains the efficacy of recovering the LVs from filter estimates. On the other hand, the results for Lorenz-96 suggest that most of the vectors, except the most stable and most unstable, are highly sensitive to the

perturbations for higher-dimensional dynamical systems. This is consistent with a very similar conclusion for LVs obtained from the filter estimates. In addition, using Lorenz-96 in different dimensions, we find that this sensitivity grows with the number of dimensions.

The Lyapunov vectors span nested subspaces called Oseledets' spaces. Thus, even in cases where the individual Lyapunov vectors recovered from a perturbed trajectory are not a good approximation of the true LVs, we investigate whether the Os-

475 eledets' subspaces are approximated well. In order to quanify this approximation, we studied the principal angles between the recovered and the exact Oseledets' subspaces. Our results suggest that these subspaces are less sensitive compared to the individual vectors themselves with respect to the perturbations in the trajectory. We support this claim by showing that the principal angles between random subspaces are significantly larger than those between the recovered and exact Oseledets' spaces.

An important direction for future research would be investigating the sensitivity of the Lyapunov vectors in different con-

480 tracting and expanding regions of the phase space. This may better capture the local sensitivity accounting for the variations in the local stable and unstable subspaces over different points on the attractor of the system. Extending the analysis discussed in this work to the case when model errors are present is another interesting direction for future work. Applying a similar analysis



to PDEs and high-dimensional models with multi-scale dynamics and spatial structures would be highly relevant to practical problems such at those in earth sciences.

*Code availability.* We provide the codes required for all the computations related to different models used in this paper through the following link https://zenodo.org/record/8396549.

*Author contributions.* SKR and AA contributed equally in conceptualization and methodology. SKR developed the codes and conducted the numerical experiments. Both the authors contributed equally in the analysis and writing the manuscript.

*Competing interests.* AA is a member of the editorial board of Nonlinear Processes in Geophysics. The authors declare that they have no
conflict of interest.

*Acknowledgements.* The authors acknowledge the support of the Department of Atomic Energy, Government of India, under project no.RTI4001.



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
