# Peer review of "Computation of covariant lyapunov vectors using data assimilation"

_EGUsphere, 2023_

## Author Comment (AC1)

**Responses to referee comments**

We sincerely thank both the referees for the detailed and thoughtful comments. All these comments helped us tremendously to improve the manuscript, which we hope will be found suitable for publication.

At the outset, we completely agree with one general theme of these comments which is that the algorithm used is by itself not new but a modification of the algorithm of Ginelli et al., with appropriate changes required to deal with the fact that data assimilation (at least EnKF) leads to trajectories with discontinuities. We also agree with the other general theme that there are multiple important questions that arise naturally out of this work, such as (i) studying larger dimensional / more realistic models, (ii) effects of model error, (iii) a detailed explanation of the results, and others. We indeed intend to pursue these in future. But we think that this paper takes an important first step which is the unequivocal and clear demonstration of the sensitivity of the LVs and the associated Oseledets' spaces in the simplest of settings with "classical" models – Lorenz-63 and Lorenz-96: a result that is not obvious from existing mathematical or numerical work. We would like to emphasize that the investigations in this paper focus on perturbations in the space of trajectories, and not in the space of initial conditions, as has been stated in the last few sentences of section 2.4. But in order to further stress this aspect in the manuscript, we have now modified the caption of figure 1, added a sentence in the conclusions, line numbers 483, as well as a sentence at the end of first paragraph in section 2.4, line number 217.

In the rest of this document, we present the detailed responses to each of the comments (indicated by a different color) and also point to the changes in the manuscript that address them. The modified manuscript clearly shows these changes, again with a different color.

**Anonymous Referee 1 (RC1)**

The manuscript investigates the estimation of the Lyapunov exponents from noisy system trajectory. The noisy trajectory is generated in two approaches: 1) posterior analysis mean of an ensemble Kalman filter by assimilating partially observed noisy observations. 2) direct perturbation of the model trajectory at observation time without data assimilation (DA).
The manuscript presents the changes of Lyapunov vectors and Oseledets' subspace when erroneous trajectory is used compared to a reference (true) trajectory in these two scenarios. The problem is worth investigating. However, there are major concerns about the manuscript.
Author response: We thank the referee for a very careful review of the manuscript.

**Major comments**

**Comment 1:** The manuscript claims it presents a new algorithm to compute the covariance Lyapunov vectors using partial and noisy observations. I find the claim troublesome for the following reasons: 1) the method relies on an underlying model to reconstruct the full system state resulting an algorithm that is exactly the same as Ginelli et al (2013); This is equivalent to using a 'surrogate model' to compute LVs; 2) although experiments use partial and noisy observations, the error of the model itself, comes entirely from model instability arising from inaccurate initial conditions. That is, it is doubtful that the approach can behave well if the model itself is erroneous/biased. Hence, I suggest the authors remove this claim and rephrase the objectives to the differences of the LV between true trajectories and the one estimated by DA.

Author response: We agree with this comment, and a similar comment of referee 2 addressed later, that the computational algorithm used to compute LVs is Ginelli's algorithm and the main focus is to understand the comparison between the approximate and the exact LVs and their sensitivity to noise. We have modified the second sentence of the abstract and also made changes on line numbers 54 of the introduction Section 1, as well as line number 468 in the conclusions Section 5 (deleted sentence starting with "We propose..."), in order to clearly reflect this point.

**Comment 2:** I am in general in support of using first-person pronouns for scientific writing. However, I feel using 'we' in Section 2 is slightly misleading. For example, Section 2.2 is describing the algorithm

proposed by Ginelli et al (2013) and using 'we' can give readers a feeling that it is the approach proposed by the authors.

Author response: We have moved the details of the Ginelli's algorithm to the appendix. The modified section 2.2 is thus compact and only gives a broad overview of the algorithm. The hope is that this reorganization will make it clear that the algorithm is indeed a well-known one.

**Comment 3:** The results show that the estimated trajectory works better for the first and the last LVs than other LVs and the Oseledets' subspace is aligned for most dimensions. Based on Bocquet et al, 2017, the error of EnKF should converge to the unstable-neutral subspace. Similar conclusions were found by some other studies, for example, Chen, Y., Carrassi, A., and Lucarini, V.: Inferring the instability of a dynamical system from the skill of data assimilation exercises, Nonlin. Processes Geophys., 28, 633–649, `https://doi.org/10.5194/npg-28-633-2021`, 2021. ; Dan Crisan, Michael Ghil; Asymptotic behavior of the forecast–assimilation process with unstable dynamics. Chaos 1 February 2023; 33 (2): 023139. `https://doi.org/10.1063/5.0105590`; Based on this line of findings, can the authors give better explanations for your results? Is it because that when the Ginelli et al (2013) is used, the errors in the mean state are projected to stable components of the LVs as well? If the observations are not partial but just noisy, will the results change?

Author response: We thank the referee for these references. We do acknowledge that a clear explanation of the result that only a few of the LVs are approximated well while the others are not approximated well is still missing. We do think that an explanation along the lines proposed by the referee may be feasible for finite-time Lyapunov vectors and possibly may be studies using the appropriate projections along the stable and unstable directions as suggested above, but this approach will be highly non-trivial for the LVs which are asymptotic in time. We feel that such detailed explanation, both numerical and theoretical, is beyond the scope of the current work, but certainly an excellent avenue to be pursued in future. With regard to the second question, the fully observed case will not be qualitatively different since these will be similar to the results with perturbed trajectories. We have added a sentence on line number 494 in conclusions Section 5 to address this comment.

**Comment 4:** I feel I don't fully understand the principal angle metric used by the authors. Considering that BLVs are orthonormal, based on Eq 14, is the principal angle just a reorder of the angle between BLVs? Can authors give a better explanation for this?

Author response: The principal angles are not just a reordering of the angle between the BLVs but they give the "minimum angles" between the corresponding subspaces. They capture the "extent of overlap" between the subspaces, as we have tried to explain in the paragraph before the definition of the principal angles. Just for concreteness, here is one specific example: consider two different sets of orthonormal basis vectors, e.g., $\{e_1, e_2\}$ and $\{(e_1 \pm e_2)/\sqrt{2}\}$ of a two-dimensional plane in $\mathbb{R}^d$ with $d > 2$. These two bases define the same space and the principal angles would be zero, whereas the angles between the two basis vectors are not non-zero: in this example, they are $\pi/4$.

**Minor comments**

**Comment 1:** L97, I'm not sure if we should call f a vector field as I think it is a map.

Author response: $f$ is indeed a tangent vector field, as it maps a point in phase space to a vector in the tangent space.

**Comment 2:** L147, I feel the statement that 'the dimensions of i-th forward and backward Oseledets subspace $S_i^+$ and $S_i^-$ are, . . ., is $n + d_i$' needs some revision. Based on Eq. (5), $dim(S_i^+) = n$, and if $dim(S_i^+) = d_i$, the sum of these dimensions must be $> n + d_i$. That said, I think the symbol $d_i$ is not defined clearly. This is only clear once $d_i$ is, for example, for the dimension of $S_i^+ \setminus S_{i+1}^+$, I think.

Author response: We thank the referee for pointing this out. We have now added the phrase that $d_i$ is the dimension of $S_i^+ \setminus S_{i+1}^+$, which is also the degeneracy of $\lambda_i$, just after and before equation (5), and also mentioned relation to dimensions of $S_i^-$ spaces just below equation (6). With this definition, we confirm that the dimensions mentioned in line 161 in the paragraph below equation (8) are indeed correct.

**Comment 3:** In Fig 1, I think there is no inverse sign in $B_j + lR_j, j+1$

Author response: We thank the referee for noting this. We now have replaced figure 1 with the correct expression which does not include the inverse.

**Comment 4:** Equation 11 can be better explained by saying D is local growth rate.

Author response: We thank the referee for pointing this out. We have now explicitly stated this in line number 521, right after equation (A3).

**Comment 5:** 5. L225, 'This leads to a our proposed modification ....' I don't think 'a' is needed.

Author response: We have now removed 'a' from the sentence.

**Comment 6:** L228, 'with $B_j$ as the initial condition for (2)'; When you compute CLVs, do you use a new matrix or is $B_j$ simply the ensemble perturbations in the EnKF?

Author response: $B_j$ is the solution of equation (2) at time $t_j$. It is not a new matrix, neither is it constructed using the EnKF perturbations. Essentially, the matrix valued solution of equation (2) has continuous trajectories, even when using the base trajectory which has discontinuities at observation times.

**Comment 7:** L234, isn't $l_2$ and RMSE the same in this context?

Author response: The distinction we had in mind was that $l_2$ error is a function of time and the RMSE which is averaged over time. Since we focus on RMSE, we have removed the reference to $l_2$ error.

**Comment 8:** Eq. 14, what is $y_k$ here?

Author response: We thank the referee for pointing this out. We have rectified this in the new equation (10).

**Comment 9:** L327, if only y is observed, I believe $H = [0,1,0]^T$ instead of $[1,0,1]$

Author response: We thank the referee and change the observation operator H in line number 307.

**Comment 10:** L328, what is $x_0$?

Author response: $x_0$ is the initial condition used for generating the true reference trajectory. $x_0$ is obtained after a long forward integration from some random initial state, so that it is on the attractor. We have now added this detail in second sentence of section 3.2.

**Comment 11:** L343, are 25 ensemble members still being used for L96? What is the ensemble size being used? The ensemble size depends on the size of the unstable-neutral space, but not necessarily the system dimension.

Author response: We agree with the referee about the size of the ensemble. In fact, that is the main reason that we use a large ensemble with 25 ensemble members for L96 in order to avoid the fine tuning of inflation and localization which may be required for smaller ensemble sizes.

**Comment 12:** Is it possible to give the RMSEs for each variable in L63 model?

Author response: The RMSE for the whole vector is shown for each different value of observational noise $\mu$ in figure 2 (on top of the line). Since the RMSE values for each component are similar, we do not report them in the paper. As an example, for $\mu = 0.3$, total $RMSE \approx 0.28$ while the component-wise RMSE are indeed close to $0.28/\sqrt{3}$: $RMSE_x \approx 0.11, RMSE_y \approx 0.15, RMSE_z \approx 0.2$.

Author response: We thank the referee for pointing this out. We have corrected this mistake.
* * *
**Anonymous referee 2 (RC2)**

This manuscript investigates the sensitivity of Lyapunov vectors computation (using the QR and Ginelli algorithms) with respect to perturbations introduced along trajectories of chaotic dynamical systems. These algorithms are used respectively to compute Backward Lyapunov Vectors (BLVs) and Covariant Lyapunov Vectors (CLVs), and the present work studies how the sought Lyapunov vectors and Osedelets subspaces are impacted by the perturbations.
The perturbations considered are of two kinds:
- obtained through an ensemble Kalman filter, filtering observational errors introduced with respect to a reference trajectory.
- obtained by perturbing directly the said reference trajectory by Gaussian white noise.
This research question is interesting and has many "real world" applications, however the present analysis is rather shallow, the methodology is inadequate, and the presentation of the results suffers from many problems. Before explaining all of this in more detail, I must say directly that my recommendation is to do a very major revision of the paper, which is not suitable for publication in the present state.
The manuscript is actually at the limit of needing a complete resubmission, and it is up to the authors to reflect on whether this is needed.
At the end of this report I propose some further studies and improvements which could lead to a suitable manuscript for the next revision, at least in my opinion.

Author response: We sincerely thank the referee for their extremely thorough and insightful comments. We have addressed the major and the minor comments below and also included additional changes in order to make the manuscript suitable for publication in our opinion.

**General comments**

**Comment 1:** In general the manuscript was quite difficult to read, and I had to go back and forth constantly (a bit of "back and forth" is ok of course).

Author response: We have rewritten and rearranged major parts of the manuscript, in particular section 2 and 3, in order to improve the flow of the manuscript.

**Title and abstract**

**Comment 1:** The title of the manuscript itself is misleading since it tends to indicate that this is a new method to compute LVs, while in fact state-of-the-art methods are used and simple sensitivity analysis about them are performed. Something like "Sensitivity analysis of Lyapunov Vectors computation with respect to perturbations" would be more accurate.

Author response: The main motivation for the study is to understand how well the LVs computed using trajectories obtained by data assimilation approximate the LVs of the true observed trajectory. The sensitivity analysis of the approximate LVs is a necessary step in such a study. Our intention behind the title was not to indicate development of a new method, but arguably that is a potential interpretation! In any case, we have now changed the title to "A note on sensitivity of Lyapunov vectors" - even though it obscures the original motivation!

**Comment 2:** Again, in the abstract, it is stated that
"We propose a method using data assimilation to approximate the Lyapunov vectors using the estimate of

the underlying trajectory obtained from the filter mean."
and this is misleading since the method to compute the LVs are the well-known and unmodified QR and Ginelli algorithms, the core of the study here being the sensitivity analysis. Thus this abstract needs to be clearer about what the subject of the article really is.

Author response: In continuation to the response to the previous comment, we have now modified the abstract to address the above concerns.

**Introduction**

**Comment 1:** In the introduction, it is asserted that the used trajectories are not shadowing trajectories. Could you please cite some works exploring this aspect in data assimilation (DA)? As stated, the noise is injected at discrete observation times, therefore in between the evolution is deterministic. How can you conclude that the portion of trajectories in between observation times are not shadowing?

Author response: The relation to shadowing is not a major focus of this paper and we have not conducted an extensive study to decide whether the trajectories are shadowing or not. Thus, we have removed the reference to shadowing from line number 86.

**Section 2**

**Comment 1:** Section 2.1 : You do not mention or explain the problem of degeneracies of the Lyapunov exponents. Also, related to that, please define clearly the dimension $d_i$ of the Osedelets subspaces. This may help the reader understand the dimension of the intersection of the Osedelets subspaces.

Author response: It is true that there may be cases where due to degeneracy, the corresponding Lyapunov vectors are not unique but the subspaces are. We have now clearly defined $d_i$, the degeneracy of the distinct lyapunov exponents $\lambda_j$ just before and after equation (5), in line number 124. Thus, with $s$ distinct exponents, $n = \sum_{j=1}^{s} d_j$, $s \leq n$, we have $dim(S_i^+) = \sum_{j=i}^{s} d_j$, $dim(S_i^-) = \sum_{j=1}^{i} d_j$ such that the sum of their dimensions is indeed $n + d_i$, as explained in the last paragraph of section 2.1.

**Comment 2:** Line 138: "see the review in Kuptsov..."

Author response: We have made this correction in the manuscript.

**Comment 3:** Line 146: "only the latter of the two limits above."

Author response: We have now made this correction in the manuscript in line number 159.

**Comment 4:** Eqs. 5 and 6: it should not be the Greek letter phi but the non-empty symbol instead.

Author response: We have now made this correction in those equations.

**Comment 5:** Section 2.2: This is a rather tedious section to explain an already well-documented algorithm in the literature. Therefore I would move most of this part to an appendix, keeping only a qualitative sketch in order to explain the following experiments.

Author response: We thank the referee for this valuable suggestion. We have modified section 2.2 by keeping only qualitative description of the algorithm and moved the rest of the details to the appendix in order to improve the flow of the manuscript.

**Comment 6:** Section 2.2: Please use capital letters for "Gram-Schmidt".

Author response: We have now made this correction in the manuscript.

**Comment 7:** Section 2.4: A general problem with the methodology of the study is that it is difficult to grasp what the different amplitudes of noise used to perturb the trajectories represent with respect to the typical variability of the model. Also, using the same amplitude for each variable - assuming they have the same variability - may bias the results. One way to solve this would be to express the perturbation in each variable as a percentage of its variability, and then vary this percentage, and not the absolute noise amplitude.

Author response: We agree with the referee and now include the comparison of the perturbation amplitudes for the different cases with the climatological variability of the corresponding system in section 3.2, around line numbers 308 and 331. Our choice for using the same amplitude for different components is because the different components have similar climatological variance. The choices for the amplitudes for the perturbations were made in order to make the perturbed and assimilated trajectories comparable in terms of their RMSE.

**Comment 8:** Section 2.4: Also, I would move section 2.4 to after section 3.2, since this one is more about the study methodology than about the methods used. (I got lost looking for the details about the perturbations while reading the results section 4.2.).

Author response: We have modified section 2.4 and section 3.2 accordingly, keeping the details of the perturbation study methodology to section 3.2 . The content in section 2.4 which pertains to the description of the methods is retaining methods part of section 2.4 in the manuscript.

**Comment 9:** Section 2.4 : There is a risk of confusion between the $x_k, y_k$ ... and the symbol used in the model's equations $(x, y, z)$. Similarly, sigma can represent the noise amplitude or is a parameter of the L63 model. Notations need thus to be revisited over the whole manuscript.

Author response: We have now introduced new notations in equation (15) and (16) in order to avoid this confusion. Accordingly, we have updated the figures with the revised symbols through out the manuscript.

**Comment 10:** Eq. 13 : Please specify explicitly the symbols meaning, like the distribution you are using and $I_d$.

Author response: We have now included the complete description of the distribution and other symbols used in equation 13.

**Comment 11:** Line 246 : Why do you use bold I for the identity here, it is the only place where it is noted this way.

Author response: We have now modified this in the manuscript in the line below equation (9).

**Comment 12:** Eq. 14 : $y_k$ should be $q_k$

Author response: We have now modified the equation with the correct expression.

**Comment 13:** Lines $293 - 297$: It should be specified that subspaces hence constructed approximate the Osedelet subspaces.

Author response: Line number 271 has been modified to make this point explicit.

**Section 3**

**Comment 1:** Section 3.2: Is that correct that you assimilate every Delta t timeunit? If so, you should specify it, by saying that you observe AND assimilate.

Author response: We have now rewritten many of the details in section 3.2 about various hyperparameters, including the time interval between observations etc., in to bring more clarity.

**Comment 2:** Section 3.2: Again, here, mu should be expressed as percentage of the variability of the concerned variables, otherwise it is difficult to estimate what this quantity means with respect to the system at hand.

Author response: We thank the referee for this suggestion. We have added this detail for better comparison in line number 308 and 331 in section 3.2.

**Comment 3:** Line 327: With respect to equation 15 order, H should be $[0, 1, 0]$.

Author response: We have made this change.

**Section 4**

**Comment 1:** Figure 2: There is a problem with the left panel. BLV 2 curve should be BLV 1 (compare with right panel CLV 1, it should be CLV 1 = BLV 1). Actual results for BLV 2 seem to be missing. Also the RMSE of the analysis displayed on top of the plots as a scale is a very bad practice, please find another way to represent it (for example you can put the numbers on top of higher 75th percentile).

Author response: We thank the referee for bringing up this point. BLV 1 and BLV2 plots overlap in this case, where as CLV 1 and CLV 2 plots are distinct. We have now changed the figure to present the data in the same fashion as for the Lorenz-96 model and only mentioned the RMSE in the figure caption.

**Comment 2:** Line 370 to 372: The phrase "We first note that since BLV are orthonormal, two of these angles are necessarily equal which happen to be those between the second and third BLV and they are quite small even for the largest observational noise strength we have used." is difficult to understand if not false. Figure 2 shows the angles between reference and perturbed vectors, not the angle between vectors of different rank. This sentence needs to be rephrased. Also what does "quite small" mean here? A more general comment is that the left panel of Figure 2 shows that comparing BLVs in such a constrained 3-dimensional system does not make a lot of sense. Also, in rather specific systems like L63, it is well possible that the results you obtain here are not generic, with very low sensitivity compared to other higher-dimensional models.

Author response: We now have improved our explanation by rephrasing the sentence in line number 379 properly on why BLV 1 and BLV 2 seem to overlap. This is because in 3-dimensions, given two sets of orthogonal basis vectors $\{a_i\}$ and $\{b_i\}$ with angles $\langle a_i, b_i \rangle = cos(\theta_i)$, if $\theta_1 = 0$, then, we have $\theta_2 = \theta_3$. We agree that due to this reason, the BLVs in L63 comes with the above drawback in 3-dimensions. However, we cannot comment on the genericity in this case as we also have low sensitivity of CLVs that are not an orthogonal basis unlike the BLVs.

**Comment 3:** Line 372 to 373: "In addition we note that the median of angle between the first - most unstable - BLV is also within 15 degrees and does not increase rapidly with the observation noise strength $\mu$." Again, this sentence makes sense only if BLV 2 is BLV 1 on Figure 2.

Author response: In response to the previous comments, we have now clarified that the BLV 1 and BLV 2 plots are very closed overlapped and this can be seen quite clearly in the current version of the figures.

**Comment 4:** Figure 4: It should be specified somewhere that i is a number indexing the principal angles.

Author response: We have now included the description of 'i' as a number indexing the principal angles in the new figure caption.

**Comment 5:** Figure 6: This shows why using L63 for sensitivity analysis of the stability directions of trajectories lead to non-generic very low sensitivity, with large contiguous regions of the attractor where the stability changes smoothly and slowly. In my view this is why studying L63 in the present framework is not very informative.

Author response: We agree with the referee that proposed explanation may indicate that the Lorenz-63 results are non-generic. Indeed, due to the lack of any mathematical backing, it is difficult to guess about genericity of sensitivity results.

**Comment 6:** Figure 7: Suffer from the same problem as Figure 2 with BLV1 being represented as BLV2, and actual BLV 2 is missing. Also, the x axis scale is wrong. Also, it is not specified what the insets are representing.

Author response: Figure 7 has now been modified to make the overlap for BLV 1 and BLV 2 plots clear. We have also included the description of the inset in the figure caption.

**Comment 7:** Line 412-413: "The 1 st and 2 nd BLV have the same rate, whereas the rates are different for the 1st and 2nd CLVs." It is impossible to verify with the BLV 2 results missing.

Author response: The modifications in figure 7 now support this statement about BLV 1 and BLV 2.

**Comment 8:** Line 413-415: The plot of the exponents seems to be missing. Therefore it is difficult to understand this statement about the order 0.2 of the exponents.

Author response: We have included the missing plot in the right panel of figure 7.

**Comment 9:** Lines 429-432: It is concluded that the Lyapunov spectrum is quite robust to perturbations of the trajectory, but this is not surprising since there is no model error and the Lyapunov spectrum is a global property of the system. What would have been more interesting to see here is the impact of the perturbations on the local exponents.

Author response: We agree with the referee that it is not surprising that the spectrum of the global exponents not being significantly affected by the perturbations. The impact on local exponents would be very interesting to study and we have added a comment in section 5 'conclusions' on line 491 about exploring this direction of future research.

**Comment 10:** Line 434-435: 'instead of the individual vectors' what does it mean? Please rephrase.

Author response: We have removed the phrase from line number 445.

**Comment 11:** line 445: 'ture' should be 'true'

Author response: We have now rectified this mistake.

**Comment 12:** Figure 9: Again, i should be identified as the index of the principal angle. Also, I would collapse both panels into one figure for a proper comparison.

Author response: We have the description in the figure 9 caption about '$i$' being the $i^{th}$ principal angle. We find that collapsing the two panels leads to a lot of overlapping line plots, degrading the clarity and readability of the two plots. Hence we have retained the two figures as originally presented in the manuscript.

**Conclusion**

**Comment 1:** Line 475: 'quanify' $\rightarrow$ 'qualify'

Author response: We have now rectified this mistake in line number 487.

RC2: Beside the problems of the manuscript and of the figures (but which could be improved), I recommend a complete major revision of the paper for the following reasons.

**Comment 1:** The study is not comprehensive enough

The only message that the paper finally conveys is that the Osedelets subspaces spanned by the BLVs are less sensitive to perturbation than particular stability directions, which can intuitively be understood as "it is harder to perturb volumes than directions". Therefore the results obtained in this paper are not surprising. More precisely it is stated starting line 395 that:
"embedded within any high dimensional Oseledets subspace, there are lower dimensional subspaces which are close to the true [BLVs] subspaces. In order to understand this behaviour more clearly, we now study the dependence of this approximation on the strength of perturbations of the trajectories."
The ensuing perturbation study does not give a clue on the characterization of the said embedded subspaces but rather confirms that perturbing (thus apparently in any way) the trajectories perturb the stability directions a lot, but less the volume spanned by the BLVs. In my opinion, the study about the principal angle is not enough. This paper is not about introducing and testing a new method, so what is left is a rather simple sensitive analysis in two very idealized models which confirms and quantifies an intuition.

Author response: We agree with the referee that the question we try to address is simple to state and the methodology needed to address the question is also quite straightforward, but the results are not at all obvious or just a trivial extension of what is known about the stability of the LV.

(a) As we explained in the paper, the assimilated trajectories, e.g. the filter mean interpolated by RK4, are discontinuous, piecewise differentiable. Thus none of the previously known mathematical results about the Hölder continuity of LVs with respect to initial conditions actually apply in this case. We are also not aware of numerical results demonstrating, e.g., such Hölder continuity even for non-chaotic systems, in which case it is at least feasible to conduct such numerical study. It would be impossible to conduct such a numerical study for a chaotic system, since the trajectories diverge due to sensitivity to initial conditions! We think that the lack of even such potentially "simple" results in published literature is an indication that indeed the sensitivity results we present are not "obvious" or "trivial" but are surprising and thus non-trivial, since our results represent sensitivity in the space of trajectories and not just the space of initial conditions.

(b) It is surely true that one of the messages is the following: it is harder to perturb volumes *spanned by LVs*, even though volumes spanned by arbitrary vectors are easy to perturb, as conveyed in the right panel of figure 9. Hence this is a non-trivial result as well. Another motivation for study of the subspace angles between different dimensional subspaces is in the next point below.

(c) The other reason why these results are significant is as follows: the assimilation in unstable subspace (AUS) methodology requires knowledge of the unstable space spanned by LVs. Our results indicate that if one needs the space $S_N$ spanned by the first $N$ true LVs, then a sufficiently larger space spanned by approximate LVs (e.g. $2N$ dimensional one) contains $S_N$ within itself up to a threshold of around 5 degrees (in terms of subspace angles). This is the main message conveyed by the results related to subspace angles in the manuscript.

Thus for all the above mentioned reasons, the results in the manuscript are novel and significant. With the various changes as discussed in the responses to the other comments, we hope that the manuscript is now suitable for publication.

**Comment 2:** Methodology

What could have been interesting is to compare the two perturbation methods, because as acknowledged in the conclusion, investigating the sensitivity conditional on the stability directions of the perturbations

is interesting. But EnKF is already doing this in part, because in perfect chaotic models the analysis error covariance matrix converges to the unstable-neutral subspace. So making the two experiments more comparable could have already led to a result about the impact of the structure of the perturbations.

Author response: Comparing the two perturbation methods, if it were feasible, would be surely interesting. One of the main results of the paper is that it would be highly non-trivial to find a method to perform such a study, since computing the stability directions, more than a few of them, is highly sensitive to the *trajectory* perturbations. This is the main reason we have mentioned this as an interesting future direction of study. Amongst other directions of future studies, one could be the following: what are the principal angles between the subspace spanned by the first $k$ BLV around the EnKF mean and the first $k$ eigenvectors of the EnKF covariances, for different values of $k$? The former are approximate while the latter are potentially - at least theoretically - along the unstable directions. Indeed there is a wealth of questions to be explored!

**Way to improve and resubmit the manuscript**

**Comment 1:** I would regroup the results per model, and not per perturbation method. Since L63 is not very interesting for the purpose of the present study, therefore I would keep this part as small as possible.

Author response: As stated earlier, the main motivation is to understand how good are the LVs obtained from assimilated trajectories - this is the main reason we first present the results for this case for both the models. The natural question that follows is to investigate this sensitivity systematically, again for both the models. Arguably, grouping per model and the grouping we chose each have their pros and cons.

**Comment 2:** I would suggest to try to characterize the subspaces which remain closely aligned under perturbation. One way to do this would be to project the true BLVs or/and CLVs on the perturbed Osedelets subspaces, and compute thus the angles between the true BLVs or/and CLVs and these subspaces, but there might be other more insightful methods.

Author response: We certainly can perform this in order to compute these angles but they will not help to characterize "the subspaces that remain closely aligned under perturbations" since they do not provide the answer to the following question: given $2N$ perturbed / approximate BLVs, how does one identify the smaller $N$ dimensional *true* Oseledets' subspaces that is hidden within it, *without* having access to the true BLVs? Projecting the true BLVs on the perturbed Oseledets subspaces will only reaffirm that such a space exists.

**Comment 3:** I would investigate what happens if perturbations are done in particular stability directions/subspaces.

Author response: As we discuss in the response to "Comment 2: Methodology" just above, such a study would be interesting but is not performed due do significant difficulties discussed in detail in that response. It is our belief that the main result of this paper, that most LV except few are very sensitive to perturbation, will also hold for perturbations in particular directions, but such a guess can only be confirmed or refuted by more numerical studies, which need to be taken up in future.

**Comment 4:** I would also make the two already performed experiments more comparable (and do this comparison per model), such that a comparison between Figures 4 and 9 is possible.

Author response: We agree with the referee that such a comparison is meaningful and in fact, the two experiments *are* comparable in the following exact sense: the perturbations of the trajectories were chosen to be comparable to the RMSE of the assimilated trajectories. We have clarified this point explicitly in the following places: line number 317 for Lorenz-63 and line number 338 for Lorenz-96 models.

**Comment 5:** I would also investigate other models, closer to geophysical applications, to see if some genericity of the results can be found.

Author response: As mentioned at the very beginning of the response, we agree with the reviewer that investigating other models would indeed be an interesting and necessary next step, and models closer to geophysical applications would be one such choice. In fact, investigating the sensitivity of non-chaotic systems, as well as that of piecewise continuous dynamical systems (for which even the original trajectory – and not just the assimilated or perturbed trajectory – would be discontinuous) would be two other interesting directions, among quite a few others, and these would help address the question of genericity within appropriate class of dynamical systems. We do hope these studies are indeed taken up, and we are ourselves interested in continuing to investigate some of them.

But we strongly feel that, as argued throughout these responses, the current work presents a very clear and unequivocal demonstration of the questions involved in addition to presenting significant results about sensitivity of LV to *trajectory perturbations* and offering clear paths for future investigations.

RC2: Because this might require a lot of additional work, I recommend at least a very major revision.

Author response: We certainly agree wholeheartedly with the referee that investigating the many interesting questions raised by this study would require quite a lot of additional work, and we feel that such future studies will merit entirely distinct and new manuscripts, on which we will be glad to collaborate with the referee or their group! :)